# Service quality, satisfaction, and intention to use Pourasava Digital Center in Bangladesh: The moderating effect of citizen participation

Bikram Biswas[1,5], Mohammad Nur Ullah[2]*, Md Mostafizur Rahman[3,6], Anas Al Masud[4]

**1** College of Public Administration, Huazhong University of Science and Technology, Wuhan, China, **2** Department of Public Administration, Bangladesh University of Professionals (BUP), Dhaka, Bangladesh, **3** School of Management, Huazhong University of Science and Technology, Wuhan, China, **4** Department of Public Administration, Bangladesh University of Professionals (BUP), Dhaka, Bangladesh, **5** Department of Educational Administration, Noakhali Science and Technology University, Noakhali, Bangladesh, **6** Department of Management, Hajee Mohammad Danesh Science and Technology, Dinajpur, Bangladesh

* nurullahniaz@gmail.com

**Data Availability Statement:** All relevant data are within the manuscript and its Supporting Information files.

**Funding:** The author(s) received no specific funding for this work.

## Abstract

This study describes how, as part of the administrative reform of Bangladesh, most of the urban local governments have set up some public service center like Pourasava Digital Center (PDC), where ICT (Information and Communication Technology) has been commonly applied to make e-services more convenient, efficient and transparent. The current study measures the Service Quality Satisfaction and Continuous Use Intention to use Pourasava Digital Center (PDC) in Bangladesh by adopting citizen participation as a moderator. Theoretically, this study has used the DeLone & McLean Information Systems (D&M IS) Success Model and Zhang's two-dimensional satisfaction model. However, most of the existing studies in Bangladesh are qualitative, and the relationship between service quality and citizen satisfaction has not been tested. A survey was conducted based on a structured questionnaire method and data collected from 332 respondents from 05 PDC and applying structural equation modelling in AMOS software while analyzing the data. The empirical results showed that the data fit the model. The finding of this study is that information quality affects specific satisfaction but not accumulative satisfaction, and specific satisfaction might not lead to accumulative satisfaction. One of the worthy findings of this study is that citizen satisfaction is highly dependent on system quality and service quality rather than information quality. The continuous use intention of the citizen is not based on specific satisfaction but significantly depends on accumulative satisfaction. To ensure the improvement of PDC's service quality, all dimensions related to the quality of service should be modified, and the administrative system and citizens should be encouraged to participate in all aspects of services.

## 1. Introduction

A new public service management paradigm, "e-governance", has been introduced to enhance efficiency and effectiveness and make public service accessible to ordinary people [1–3]. The

**Competing interests:** The authors have declared that no competing interests exist.

public sector gets reenergized globally to enhance the quality of governmental operations by facilitating enterprises that are increasingly concerned with governments, are customer-centred, cost-effective, and give services to citizens conveniently. Therefore, e-governance and new public management are essential to reform, modernization, and government advancement [4, 5]. The evolution of public administration has been highlighted in Woodrow Wilson's famous article "The Study of Administration", published in 1887, employing paradigms that offer a solid foundation for dynamic development. Then, in 1975, Nicholas Henry introduced the five public administration paradigms [6]. This innovative approach to general management has opened up new possibilities and significantly increased its ability to provide essential public services [7–9]. The New Public Management (NPM) system has implemented theoretical public administration concepts while highlighting its ties to markets and private sector management.

Nevertheless, to some extent, e-governance repeats NPM principles and follows them in the service management system [10]. The New Public Service is a contemporary paradigm for citizen-focused public administration that challenges the 21st-century principles of new general management. New shared services emphasize democratic governance and serve citizens by empowering them strategically [11, 12]. The reflections of the e-governance and new public service paradigms can be seen in Bangladesh's PDC, which considers empowering citizens and accessing government service fast, effective, and transparent [13, 14]. The Government of Bangladesh has taken public service to the next level by setting up the Union Information Service Center (UISC) and Pourasava Digital Centre (PDC) to modernize government administration to provide citizens with more efficient and effective services [15]. Thus, it has opened the visibility of Digital Bangladesh prospects.

The Pourasava Digital Centre (PDC) is a Public Private Partnership (PPP) project to decentralize service delivery, strengthen local government, and empower communities. The project is funded by UNDP and USAID and acted under the Access to Information (A2I) Program to support establishing a digital nation by bringing digital services to the people's doors [16, 17]. The PDC has been created to meet the public service needs of the residents of Pourasava and to act as a one-stop shop for e-service delivery [18]. It provides electronic services to individuals, such as online birth registration, death certificates, citizenship certificates, computer and English language training, photocopying, scanning, email, internet surfing, mobile banking, and so on [16–19].

Being a developed country status aspirant, Bangladesh is struggling with its aged long ill-functioning public sector where it has maintained a legacy of poor citizen's expectations in its public service delivery system [20]. Challenging the prevailing context with the agenda of LDC graduation and aspiring smooth transition towards the Digital Bangladesh manifesto, the government of Bangladesh has taken a number of initiatives- mostly are ICT-based digital centers [21]. Municipality based- Pourasava Digital Centers (PDC) is one of such but less familiar than that of the UDC, the Union Digital Centers owing to less research exposure. Assessing the issues of integrating e-governance mechanisms with ICT-enabled initiatives, it has been marked that- Bangladesh is not yet up to the mark indeed [22]. Here, the proper elevation of the context is only possible with the all-round necessity assessment for issues like–IT physical infrastructure, access to internet connection, skilled workforce, e-literacy, and administrative consent from both semi-urban (Pourasava) and rural contexts. Al though, the prevailing studies has tried to address the rural dynamics, the context of semi-urban, townships are always left behind. There is a demographic difference between rural and rest which is a prevailing case too, the digital divide, e-literacy concerns are not same [23, 24]. Therefore, this study sets its trajectory towards learning the unknown semi urban dynamics of digital services in Bangladesh by the means of measuring the service quality, citizen's satisfaction, and their intention to

use the Pourasava Digital Center. Addressing the mentioned objective, the specific objectives of this study are: O1. To know the current status of service quality at PDC- Pourasava Digital Center in Bangladesh; O2. To analyze to what extent citizens are satisfied after receiving services at PDC and O3. To realize the citizen's participatory role in service quality and satisfaction.

## 2. Literature review

Over the last few decades, there has been a growing consensus among governments across the world in utilizing the rapid growth of digital technologies in the way public services are delivered. Traces can be found in developing countries, especially in Bangladesh where the public administration is making efforts to facilitate customer-centered, cost-efficient, and user-friendly delivery of public services to its citizens, thereby improving the quality of governmental functions [25, 26]. Here, the Pourasava Digital Centers (PDCs) have emerged in the country as exclusive hubs for citizens to access various government services at the doorstep. This research explores the service quality, citizen satisfaction nexus while putting a light on the intentions to use PDC in the context of Bangladesh. This research further delves into the role of citizen participation with an aim of proving a comprehensive understanding of the factors which are influencing the effectiveness of the digital-bound services in the country. In the field of digital public services and e-governance, a plethora of literature has been found. But rarely any works focused on PDC and its unique challenges and opportunities in the context of developing countries indeed.

### 2.1 Quality, service quality, and the public sector

Quality primarily refers to the degree of standard or measurement of a product or service, where it stands for assessing the core characteristics of the product or service so that it can be compared to other similar ones. The International Organization for Standardization (ISO) does not define quality as a specific concept or measurement but emphasizes the idea as the degree to which a set of intrinsic qualities of an object meets requirements relating quality management and quality assurance. Academically, though the concept varies on the sectorial and orientational basis, in general terms, "Quality" involves only the fundamental characteristics that enable services to meet explicit or implied requirements. Practically, the concept of quality requires both perspectives- the customer and the product service. Crosby [27] takes a dual perspective, defining quality as conformance to requirements from both the product and customer's perspective. Therefore, the realization of quality seeks a context of assumptions where the customer accepts the service or product as if it can satisfy a need. Despite the varied sectorial interpretations, quality generally revolved around meeting explicit or implicit requirements in general. Broh's [14] inclusion of user satisfaction added that, Quality no longer serves the requirement-based comparisons; in fact- it is the degree of excellence at an acceptable price and the controlling of variation at an appropriate rate. As the core concept of consumer expectations underpins all quality requirements, quality stands as both the feature of a product or service supplied to a consumer and the mark of a company that has pleased all its customers. In the realm of services, particularly in the public sector, attention has intensified due to the simultaneous creation and consumption of services. Scholars like Fetra and Pradiani [28], Tabaeeian et al. [29], Parasuraman et al. [30] and Zeithaml and Bitner [31] view service quality as a global judgment influencing overall customer satisfaction. Kao and Kao [32] and Beatson et al. [33] extend this perspective, illustrating the impact of service quality on employee loyalty and satisfaction, thereby influencing the perceived quality of the product or service.

Explicitly referring to the public sector service quality, Herdiansyah and Perdana [34] and Gowan et al. [35] highlighted the multifaceted nature of general sectoral service provision, involving not only meeting declared requirements but also identifying unmet needs, establishing priorities, distributing wealth, and being publicly accountable. This complexity is further compounded by the paradigm shift in public administration, as noted by Neal et al. [36] and Caron and Giauque [37], suggesting that the dynamics of public sector and dimensions of public service quality are continually evolving. These observations underscore the challenges public sector professionals face in delivering quality services amidst shifting landscapes.

## 2.2 E-service quality and satisfaction attributes

Linking perceived quality of e-services and the overall satisfaction, studies have shown a few influential factors which includes- citizen's attitude of usage [38], trust and loyalty [39], pattern and structure of a website and its information updating system, privacy, and fulfilment [40], public participation opportunities [41] and the improved socio-economic condition [42]. Web service quality does not directly affect customer satisfaction but acts as a mediator, Udo et al. [43] stated. Zehir and Narcıkara [44] argued that service quality and loyalty intention are correlated strongly, where perceived value acts as a mediator. Alsuwaidi and Sultan's [45] examination of Abu Dhabi's e-service system identifies knowledge sharing, internal collaboration, and staff motivation as pivotal factors influencing service quality. In the context of Bangladesh, behavioral intention to use service centers is stimulated by service and information quality, performance expectancy, satisfaction, and engagement in digital service delivery systems, which is a sorry case in Bangladesh indeed. Thereby, Saha [46] recommended ensuring service quality and enhancing the capacity of local e-service centers. Linking service satisfaction and quality dynamics- Biswas and Roy [15] found that satisfaction significantly depends on the quality of service, system, and information when citizen participation is a significant moderator. Intersecting the service quality and satisfaction, Hoque [47] stated that the digital divide and sound governing system significantly impacts where Khadiza and Nur Ullah [48] reported that people's satisfaction and perception shape the service quality indeed. Additionally, Anam [49] added that citizen satisfaction moderately impacts their perception of the service quality and intention to use. However, Alam [50] argued a different perspective and stated that user satisfaction solely depends on the tangible facilities, such as- supplier's responsiveness and the infrastructures. Ferdous et al. [51] claimed that the trust and satisfaction of the digital services of local governments heavily depend on the quality they assure. However, a few studies have been conducted to measure service quality, satisfaction, and user intention to use the PDC in Bangladesh. Thus, this study minimizes this research gap and contributes to generating new knowledge in this area, which simultaneously brings some arguments for future researchers.

## 2.3 Citizen satisfaction

The term "satisfaction," originating from Locke's [52] is used in measuring job performance, denotes a positive emotional thought. Anderson and Sullivan [53]; Oliver [54, 55], extend this concept to encompass a mental state reflecting one's evaluation of an organization's service quality after availing it. Bhattacharjee and Premkumar [56] and Thong and Yap [57] defined citizen satisfaction as an affection towards a service indicating the importance of public sentiments with it. Citizen Satisfaction, according to Zhang [58] comprises of two dimensions: i.e. specific satisfaction and accumulative satisfaction. This dual perspective aids in comprehending citizens' needs and aspirations, contributing to the formulation of people-oriented public policies. Giese and Cote [59] characterize citizen satisfaction as a summary affective response which varies with time and consumption of services. Carlson and O'Cass [60] found that

citizen satisfaction comes from positive service consumption. However, Brown & Coulter [61] claimed that citizen satisfaction depends on the people's demographic characteristics indeed. The observations of Morgeson's [62] further claimed that satisfaction leads peace and happiness, pushing service recipient's intention to use whereas, dissatisfaction leads to the intentions of seeking alternatives. Subramanian et al. [63] further delved into the factors influencing citizen satisfaction, i.e. service quality reliability and purchasing e-service experience.

## 2.4 Citizen participation and its moderating effects

Citizen Participation refers to the concept of engaging citizens into different state affairs and its decision-making systems to serve the people's interest, ensuring cumulative development indeed [64, 65]. Claycomb et al. [65] found citizen participation to be similar as the concept of "citizen engagement" which reduces- cost, ensures psychological satisfaction, and creates service value. Citizen participation in e-governance refers to an active interaction of the citizen while receiving services [15]. In this research, citizen participation works as a moderator, which refers to the active cooperation and interaction of service receivers with service providers of PDC through information, resources, responses, and time and costs. Thus, the PDCs provide satisfactory and quality services and create substantial value.

## 2.5 E-governance in Bangladesh

At the first stand, E-governance and E-government cumulatively hailed the same motive of utilizing information and communication technologies (ICTs) to advance government institutions' efficiency and offer government services online. However, Mukonza [6] stated that the definition of e-government later expanded to a wide range of contacts with citizens and enterprises and open government data for enabling governance innovations. In recent years, online services, big data, social media, mobile apps, cloud computing, and other digital advances have significantly impacted people and the government. With such usage of e-governance, Rahman [66] projected that the public administration performs more proficiently, providing better services and responding to transparency and accountability demands. Sooner, e-government became an alternative way of achieving public goals through digital means. Respectively, "SMART Bangladesh, Vision 2041" made a safe passage for Bangladesh, marching forward with the e-governance adaptation, which has a distinctive connotation with state growth. Despite several bottlenecks and limits, work on implementing e-governance in all administration sectors is underway. Many e-governance initiatives have been accomplished already, and more are in the works now. Over 120 million mobile customers and 43 million Internet subscribers in the country are already receiving benefits from e-governance in various ways, Rashid and Rahman [67] stated. Rumi et al. [68] indicated that greater digitalization surely would come with more excellent service coverage. The primary goal of Vision 2021, followed by Vision 2041, is to make more e-services available to individuals so that tasks like- "registration for academic institutions, publication of test results, registration to work abroad, registration for pilgrimage, delivery of official forms, online submission of tax returns, online tendering, online banking" which will turn convenient, inclusive, affordable indeed. However, most of these initiatives are concentrated in the rural local areas. Rahman [66] stated that the Deputy Commissioner Offices in districts and UNO offices in Upazila serve as nodal points of action for delivering services like e-passports to the huge numbered rural clients. Decentralization of services has thus been maintained with the adaptation of ICT into service-providing mechanisms. Zafarullah and Ferdous [21] agreed on the stand that telemedicine and videoconferencing for medical treatment, as well as for administrative tasks, are making seminal results. In this growth race, the establishment of approximately 5,000 Union Information Service

Canters significantly boosted e-service delivery, particularly in rural areas [16, 17]. The recent conversion of 8,000 village post offices and around 500 Upazila post offices into e-centres and the introduction of mobile money orders and postal cash cards added another flock. Although the Union Information Centers, along with the District Information Cells and the National Information Cell, are serving with groundbreaking improvements these days, there are more in the pipeline.

## 2.6 Pourasava Digital Center in Bangladesh

Pourasava Digital Centers (PDC) are one-stop service centers positioned across the country's 330 Pourasava. PDC began its operations in 2014, primarily led by UISC- the Union Information and Service Center, which started operations earlier, in 2009 at 30 UP as a pilot project [66]. The core objective behind the establishment of PDCs was to streamline information and government-private bound commercial services, leveraging the effective utilization of information and communication-based technology. PDC was predicted as it will eradicate (TVC); time-visit-cost of service recipients of different government offices [6]. Established under a PPP-public-private partnership modality, these centers were envisioned as one-stop service hubs of all government service-related information and assistance. At the operations, the PDC is managed by two self-employed (one male and one female) and motivated local entrepreneurs, with an oversight provided by a local advisory committee led by the mayor of concerned Pourasava with the technical assistance from initiatives like a2i, managed by the Prime Minister's Office [69], signifying a collaboration between central-local relations indeed. Notably, the government does not compensate any of these entrepreneurs in any way. Here, the Deputy Commissioner's Office sets certain standards for appointing entrepreneurs, emphasizing the importance of adequate computer abilities to operate the equipment. In fact, there is a practice where the entrepreneurs invest finances and in return, they receive a profit share based on the agreement reached between the respective committee members and the entrepreneurs [21].

## 3. Theoretical framework and hypotheses development

Demystifying e-service quality and satisfaction, various the researchers have proposed different theories, models and approaches indicating how service quality and satisfaction are achieved and explaining interrelation between them. Gilbert et al. [70] reviewed multiple ways and approaches justifying the growth of Service Quality measuring paradigms, i.e. performance approach, expectation-disconfirmation approach, attribute importance approach and service quality versus service satisfaction. These paradigms state the changing pattern of assessing individual and institutional performance, quality of services and satisfaction led by service quality. Although, Haywood-Farmer [71] identified three attributes through utilizing "Attribute Service Quality Model" i.e. Physical facilities and processes, People's behavior and Professional judgment as significant for analyzing service quality and citizen satisfaction; two critical models- SERVQUAL model by Parasuraman et al., (1988); modified in 1991 by Parasuraman, Berry & Zeithaml [15], and the other one- D&M IS Success Model by DeLone & McLean, are globally recognized and accepted. In respect of SERVQUAL, its acceptability in the domain of ICT service quality and satisfaction measurement is proved by a plethora of studies; amongst-Kettinger & Lee [72] and Susniene [73] utilized the model for realizing the nexus between service quality and levels of customer satisfaction. Pitt et al. [74] further explored deep and found that the model is suitable for measuring service quality, exposing quality elements of service delivery. The other one, D&M IS Success Model was developed by DeLone & McLean in 1992 through integrating the communication system theory of Shannon and Weaver (Al-Fedaghi, 2012); Information Influence Theory of Mason (1978) and, the empirical studies of

Management Information System (MIS) in 1980 [75]. Later on, Seddon and Kiew [76] brought a slight moderation into the model after verifying it with the empirical data. Consecutively, Pitt et al. [74] suggested revising the D&M IS model and provided five dimensions to measure IS. After such a plethora of rigorous observations and logical claims, Delone and McLean updated their D&M IS model 2003 [77]. Relevant scholars have been using this updated model since then. Wang and Liao [78]; Hu et al. [79]; Ding [80]; and Wangpipatwong, et al. [81] have already used this modified model and thereby proved the relevancy of D&M IS model in the current context.

In south Asian perspective, particularly in the context of Pakistan, Rahi & Ghani [82] explores the integration of the DeLone and McLean model with self-determination theory in the context of internet banking continuance intention, which likely delves into how these theoretical frameworks can be combined to better understand the factors influencing individuals' decisions to continue using e-services in the banking sector. He and his team also conducted another study to investigate how the diffusion of innovation theory influences citizens' intentions to adopt e-government services and examines the theory's impact on individuals' decisions to embrace digital government offerings, shedding light on the factors driving or inhibiting their acceptance of such services [83]. In respective of Bangladesh, Biswas and Roy [15] utilized this updated model to measure the IS SQ of UDC- Union digital centers. Through a rigorous understanding of the above significant models; this study adapts its analytical model based on the modified D&M IS Model (2003) and Zhang's Satisfaction Model (2009). These models are considered owing to their data relevancy and contextual orientation on Bangladesh for serving its research objectives.

The modified D&M IS model includes three specific attributes for measuring service quality: Information Quality (INFQ), System Quality (SYSQ) and Service Quality (SERQ). INFQ highlights the quality of the website; SYSQ indicates the reliability, usefulness, availability and adaptability of respective system; and SERQ defines the service quality and user's nexus [77]. According to Zhang (2009), satisfaction has two types: Specific Satisfaction (SPES) and Accumulative Satisfaction (ACCS). This model includes a dependent variable, Use Intention (USEI), and a moderator, Citizen Participation (CITP). Therefore, the research model for this study is as follows;

Information quality pertains to the presence of accurate, reliable, and readily accessible information within digital centers for users [70]. Biswas and Roy [15] examined UDCs and contended that information quality contributes positively to both overall and specific user satisfaction. Ming et al. [77] and Hossain et al. [84] similarly assert that citizen satisfaction hinges on the quality of information provided. These studies suggest that the quality and accessibility of information have a substantial impact on people's satisfaction levels and their inclination to utilize digital centers. Consequently, the 1st hypothesis is formulated as follows:

**H1a**: *INFQ has a significant positive impact on ACCS.*

**H1b**: *INFQ has a significant positive impact on SPES.*

According to Yanjun [64], system quality denotes the efficiency, responsiveness, and reliability of an organization's operational environment. This encompasses electronic and network systems that are swift and dependable, allowing citizens to derive maximum benefit [15]. The overall physical infrastructure and its seamless operation are observed to capture people's attention towards the service center, fostering a mutually beneficial scenario where individuals utilize facilities optimally. Hossain et al. [84] and Ferdous et al. [51] posit that user satisfaction with a service is contingent upon the quality of infrastructure and internal systems. Alam [50] also corroborates the positive correlation between system quality and user satisfaction,

indicating that an institutional setup with adequate facilities significantly contributes to user satisfaction. Therefore, the second hypothesis is crafted as follows:

**H2a:** *SYSQ has a significant positive impact on ACCS.*

**H2b:** *SYSQ has a significant positive impact on SPES.*

Rita et al. [40] stated that service quality refers to an assessment of received services, which affects ultimate satisfaction and directs the recipients whether they stay with or seek alternatives. In e-service management systems, service quality and user satisfaction correlate positively [15, 46, 47]. Morgeson [62] argued that people's satisfaction largely depends on service quality since they feel dissatisfied with less quality services. Along with other two i.e. information and system quality, service quality like quality of response, interaction, service timing, the abstract service meaning final product' quality, efficiency and effectiveness of that services and service centers, these are all have paramount importance and strongly interlinked with user's satisfaction. Hence, the third hypothesis is;

**H3a**: *SERQ has a significant positive impact on ACCS.*

**H3b:** *SERQ has a significant positive impact on SPES.*

Yanjun [64] noted that citizen participation in e-service delivery and management systems indicates an active involvement of service recipients with service providers in the entire process where a transparent exchange of information happens. Functional cooperation between them helps both service recipients to avail accurate services and service providers to understand the needs and aspirations of the citizens [33]. At the PDCs in Bangladesh, if the citizen can adequately communicate with the staff and actively provide relevant information, it will be easier for them to obtain more relevant and accurate information. Citizen also helps the digital center to give relevant and accurate information to the staff that will make the team more comfortable to understand the actual expectations of the citizens. If there is a lack of willingness to interact and not actively cooperate, it is challenging to grasp comprehensive information and necessary precautions. Different levels of citizen participation are likely to affect the correlation between the quality of service and satisfaction. Therefore, active citizen engagement affects the nexus between service quality and satisfaction. Citizen participation moderates service quality-satisfaction relations [15, 48, 65, 84]. Thus, the fourth and fifth hypotheses are;

**H4a:** *CITP moderates the relationship between INFQ and ACCS.*

**H4b:** *CITP moderates the relationship between SYSQ and ACCS.*

**H4c:** *CITP moderates the relationship between SERQ and ACCS.*

**H5a:** *CITP moderates the relationship between INFQ and SPES.*

**H5b:** *CIIT moderates the relationship between SYSQ and SPES.*

**H5c:** *CITP moderates the relationship between SERQ and SPES.*

Specific satisfaction defines the satisfaction for every transaction and specifies the gap between expected and actual services, but accumulative satisfaction refers to overall user satisfaction since the first transaction [58]. Caron and Giauque [37], Biswas and Roy [15], Ferdous et al. [51], and Alsuwaidi and Sultan [45] supported that specific satisfaction affects accumulative satisfaction. It is irrefutable that overall service satisfaction of a recipient about an organization mostly depends on the level of satisfaction of that person in every single services received from the same center in different time periods; the sum of all transaction may lead to his final satisfaction at the end. Therefore, the sixth hypothesis is;

**H6**: *SPES has a significant positive impact on ACCS.*

Chatfield & AlAnazi [39] and Biswas and Roy [15] stated that the user intention to use depends on customer satisfaction since satisfaction attracts the users to the next transaction. Saha [46] and Anam [49] found that satisfaction positively affects the user's mind to use for the next time. The studies claimed that satisfaction either specific or accumulative ensure the using the service center by the same recipient next time because, satisfaction build trust about the services and trust ultimately leads to the subsequent uses of that trustworthy service centers. In this regard, the seventh and eighth hypotheses are;

**H7:** SPES has a significant positive impact on CONI.

**H8**: *ACCS has a significant positive impact on CONI.*

A Mediator greatly influences the causal relationship between dependent and independent variables and shows the facts of that relationship and how and why they are related [1, 85]. In defining the relationship among the constructs, Specific Satisfaction (SPES) works as a mediator; on the contrary, Accumulative Satisfaction (ACCS) mediates the relationship between Specific Satisfaction and continuous intention to use e-services [77, 28, 84, 86, 87]. Thus, this study offers the following hypotheses;

**H9a**: *SPES mediates the relationship between SERQ and ACCS*

**H9b**: *SPES mediates the relationship between SYSQ and ACCS*

**H9c**: *SPES mediates the relationship between INFQ and ACCS*

**H9d**: *ACCS mediates the relationship between SPES and CONI*

## 4. Methodology

### 4.1 Sample and procedures

The study was conducted on the five PDCs as presented in Table 1 across Bangladesh through a quantitative method with primary data analysis. The fundamental reason of selecting the five different PDC is to ensure diversity and equal responses of all parts of the country along with considering the existing limitation of resources and time. Quantitative method here ensures the statistical significance of the variables and their impacts on each other mentioned in the hypotheses presented in the Fig 1. Using a convenience sampling technique, the service receivers of PDC in Bangladesh were invited to participate in the survey to collect primary data for this study. In quantitative research, convenience sampling technique provides easy access to the sources and is cost-effective and available [84, 88–90].

**Table 1. Areas and sample size (N = 332).**

| PDCs | Frequency | Percentage |
|------|-----------|------------|
| PDC 1 | 56 | 16.9 |
| PDC 2 | 67 | 20.2 |
| PDC 3 | 47 | 14.2 |
| PDC 4 | 50 | 15.1 |
| PDC 5 | 112 | 33.7 |
| **Total (N)** | **332** | **100%** |

Source: Field Data

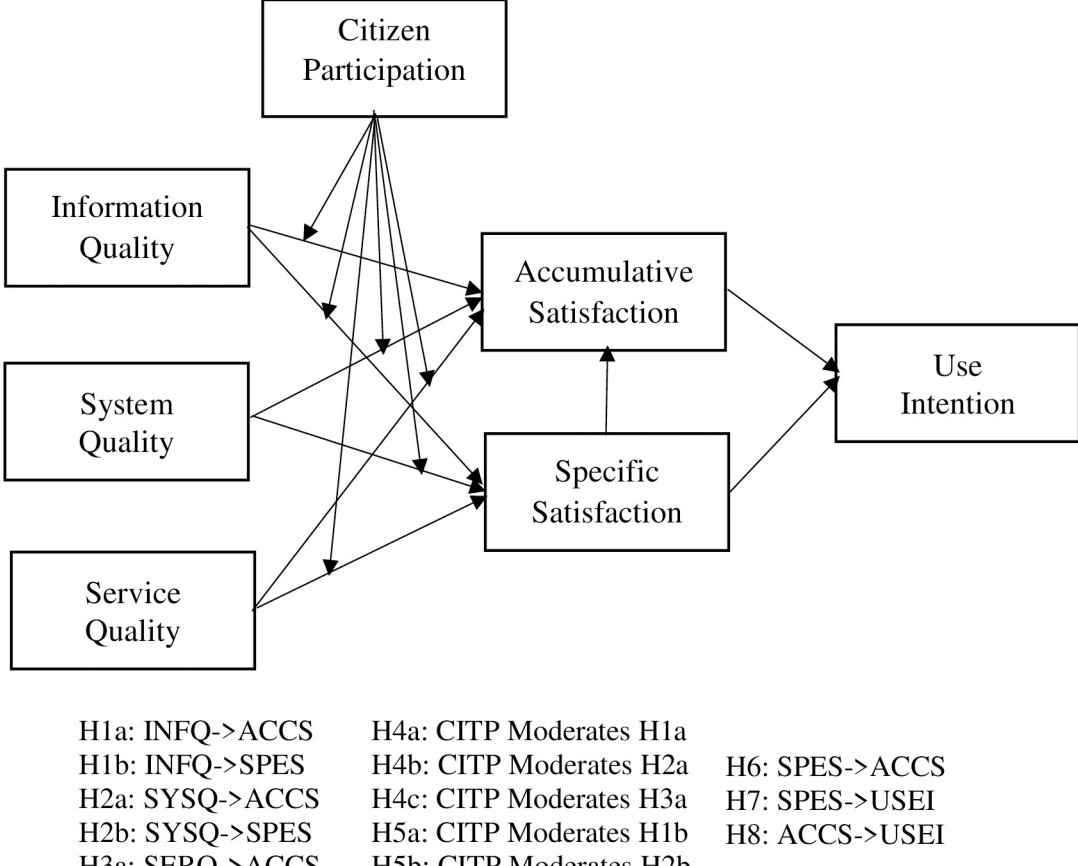

**Fig 1. Research model with hypotheses.**

Initially, some graduate students were recruited, trained and well-instructed on how to do surveys and sent to the field to collect primary data for reaching the core objectives of the study. Data collection started on May 05, 2023, and ended on June 15, 2023, just after the completion of piloting. The respondents voluntarily participated in the survey and were confirmed that their responses would be confidential and used only for research purposes. After rigorous data screening and excluding missing values and outliers, 332 valid data were used for this analysis. According to Comrey & Lee [91], the data with a sample size equal to or more than 300 is valid for analysis. The details of the sample's demographic characteristics are presented in Table 3.

## 4.2 Instrument development

The instruments and their measurements as presented in Table 2 were constructed based on the existing recent literature focusing on the context since collecting information from available sources is essential for ensuring the data reliability of any research [90]. A structured-questionnaire was developed through steps like formulating a draft questionnaire, advisory discussion, piloting and finalizing. The questionnaire was initially adopted in English and further translated into Bangla for easy understanding of the respondents. All latent variables contained in the questionnaire were measured in a five-point Likert Scale Format ranging from

**Table 2. Latent and Measured Variables.**

| Latent Variables (LV) | Measured Variables (MV) |
|---|---|
| Information Quality (INFQ) | INFQ1: PDC provides accurate information<br>INFQ2: PDC provides adequate information<br>INFQ3: PDC provides latest and updated information<br>INFQ4: PDC provides high quality information |
| System Quality (SYSQ) | SYSQ1: PDC's system responds quickly to the user's request<br>SYSQ2: PDC's System offers easy access of information<br>SYSQ3: PDC's system is of a very high quality |
| Service Quality (SERQ) | SERQ1: PDC's staffs are willing to help users and solve their problems<br>SERQ2: PDC provides safety and security for the users<br>SERQ3: PDC's staffs show positive interest in handling my affairs |
| Specific Satisfaction (SPES) | SPES1: I (user) was highly satisfied with my last transaction with PDC<br>SPES2: The last transaction with PDC exceeds my expectation<br>SPES3: I had a great experience last time with PDC |
| Accumulative Satisfaction (ACCS) | ACCS1: I always feel very good with PDC<br>ACCS2: PDC meets my expectation always<br>ACCS3: I always feel satisfied with PDC |
| Continuous Intention to Use (CONI) | CONI1: I will obviously use PDC if needs<br>CONI2: If I would choose, I could go nowhere but PDC<br>CONI3: If I could, I would like to discontinue coming to the UDC for services. |
| Citizen Participation (CITP) | CITP1: I will share my feelings with PDC's staffs about PDC's services<br>CITP2: I will provide suggestions to the PDC's staffs on PDC's services<br>CITP3: I am happy to cooperate with the PDC's staffs for related issues |

Source: Based on the Biswas and Roy [15]

Strongly Disagree (1) to (5) Strongly Agree. The questionnaire items for service quality, i.e. INFQ, SYSQ and SERQ, were taken from the studies of Wang and Liao [78], Kang and Lee [92], Mouakket [93] and Biswas and Roy [15]. The Citizen Satisfaction items like SPES and ACCS were adopted from the studies of Zhang [58]. From the studies of Biswas and Roy [15] and Zhou et al. [94], citizen participation as a moderator on quality and satisfaction was included. An item like CONI was added from the studies of Wangpipatwong et al, [81] and Yeh & Teng [95]. The Latent Variables (LV) and Measured Variables (MV) are as follows;

## 4.3 Statistical tools

This study used SPSS-20 for analyzing the descriptive statistics and calculating the validity and reliability of the items of the survey. For diagnosing Conformity Factor Analysis (CFA) and Structural Equation Modeling (SEM), SPSS AMOS-24 was used, which is widely used and accepted software for theoretical model testing and validation. In order to ensure the accuracy, suitability and precision of the parameters, these tools adopt the Partial Least Square (PLS) method. This is the best-fit tool for analyzing the validity and reliability of the measurement model and its theoretical inferences [93, 96] and for analyzing high-quality data with a large sample size [97].

## 4.4 Research paradigm

In information system research, there are essentially four paradigms that are employed: positivism, interpretive, advocacy, and pragmatism [98]. The positivist paradigm, rooted in the philosophy of natural sciences, emphasizes objectivity, empirical observation, and the pursuit of universal laws governing human behavior [99]. This study followed the positivist paradigm of information system research. In the context of researching service quality, satisfaction, and

**Table 3. Demographic features of the respondents (N = 332).**

| Characteristics | Categories | N = 332 | Percentage |
|---|---|---|---|
| Age | <20 | 22 | 6.6 |
| | 21–30 | 85 | 25.6 |
| | 31–40 | 103 | 31.0 |
| | 41–50 | 60 | 18.1 |
| | 51–60 | 47 | 14.2 |
| | >61 | 15 | 4.5 |
| Sex | Male | 191 | 57.5 |
| | Female | 141 | 42.5 |
| Education | no education | 27 | 8.1 |
| | primary | 18 | 5.4 |
| | high school | 72 | 21.7 |
| | college | 58 | 17.5 |
| | bachelor | 95 | 28.6 |
| | masters or above | 62 | 18.7 |
| Occupation | Teacher | 34 | 10.2 |
| | Farmer | 21 | 6.3 |
| | Private employee | 43 | 13.0 |
| | Business | 54 | 16.3 |
| | Housewife | 35 | 10.5 |
| | Student | 40 | 12.0 |
| | Retired | 32 | 9.6 |
| | No job | 32 | 9.6 |
| | Others | 41 | 12.3 |
| Income | <5000 | 34 | 10.2 |
| | 5001–10000 | 46 | 13.9 |
| | 10001–20000 | 73 | 22.0 |
| | 20001–40000 | 64 | 19.3 |
| | >40001 | 48 | 14.5 |
| | No income | 67 | 20.2 |
| Knowing PDC | People | 102 | 30.7 |
| | Advertisement | 96 | 28.9 |
| | PDC entrepreneurs | 40 | 12.0 |
| | Public Representative/ Govt. Officials | 65 | 19.6 |
| | Website | 29 | 8.7 |
| | People | 102 | 30.7 |
| Frequency of using PDC | 1 Time | 242 | 72.9 |
| | 2 Times | 40 | 12.0 |
| | 3 Times | 26 | 7.8 |
| | >3 Times | 24 | 7.2 |
| Service Satisfaction | Yes | 172 | 51.8 |
| | No | 160 | 48.2 |
| Service Way | Auto-help | 62 | 18.7 |
| | Staff service | 270 | 81.3 |
| Times to come for single service | One time can finish | 125 | 37.7 |
| | Need to come several times | 207 | 62.3 |

Source: Survey Data

intention to use Pourasava Digital Center in Bangladesh with a focus on the moderating effects of citizen participation, the positivist paradigm offers a structured and systematic approach. Researchers adopting the positivist paradigm would likely employ quantitative methods, such as surveys to gather data on service quality metrics, citizen satisfaction levels, intention to use digital services, and the extent of citizen participation. These methods allow for the collection of measurable and quantifiable data, facilitating statistical analysis to identify correlations and causal relationships between variables [99]. Within this paradigm, hypotheses would be formulated based on existing theories and literature, proposing specific relationships between service quality, satisfaction, intention to use, and citizen participation. The research design would aim to control for confounding variables and ensure the reliability and validity of findings [99]. Moreover, the positivist approach encourages the use of standardized instruments and systematic sampling techniques to ensure generalizability and reliability of results. By adhering to rigorous methodologies, researchers can produce objective insights into the factors influencing citizens' perceptions and behaviors regarding digital service utilization in Pourasava Digital Centers.

## 5. Results

### 5.1 Descriptive statistics

The participants profile as presented in Table 3 in this study primarily consist of individuals aged between 31 and 40 years old (31%), followed by those aged 21 to 30 (25.6%), and 41 to 50 (18.1%). The majority are male (57.5%), predominantly hold bachelor's degrees (28.6%), and a significant portion are self-employed (16.3%). The highest proportion of respondents earns between BDT 10001–20000 (22%), while 20.2% reported having no income. Information about PDC was predominantly acquired from previous service usage (30.7%) and advertisements (28.9%). Most respondents (72.9%) utilized PDC for the first time, with a nearly equal split between satisfaction (51.8%) and dissatisfaction (48.2%). A majority (81.3%) received direct staff service, while the rest (18.7%) utilized auto-help services. Despite multiple visits to PDC by 62.3% of respondents to access a single service, dissatisfaction with the service (48.2%) was noted. The demographic details of the participants are provided in Table 3.

### 5.2 Scale reliability and validity

In this study, standardized factor loading was employed to assess the reliability of the scale, with all items showing values above 0.70, as detailed in Table 4 and Fig 2. Additionally, Cronbach's Alpha and Composite Reliability (CR) were calculated to verify internal validity, while Average Variance Extracted (AVE) was used to assess convergent validity. The Cronbach's Alpha values ranged from .794 to .919, indicating high internal consistency, while CR values ranged from 0.744 to 0.933, surpassing the standard threshold of 0.70 for reliability. Similarly, AVE values ranged from 0.523 to 8.22, all exceeding the recommended threshold of 0.50 for convergent validity. These findings suggest that the measurement instruments utilized in the study are reliable and valid for assessing the constructs under investigation, in accordance with established criteria [88, 100].

In addition, the study utilized the square root of Average Variance Extracted (AVE) and the squared correlation among the constructs to assess discriminant validity. The results, presented in Table 5, indicate that the square root of AVE scores exceeds the squared correlation among the constructs, affirming the distinctiveness of the data. Furthermore, the overall model fit was evaluated using various goodness-of-fit measures, including a $\chi^2/d$ ratio of 3.07, Goodness of Fit Index (GFI) of 0.864, Adjusted Goodness of Fit Index (AGFI) of 0.812, Comparative Fit Index (CFI) of 0.943, Root Mean Square Error of Approximation (RMSEA) of

**Table 4. Reliability and validity text.**

| Constructs | Estimate | CR | AVE | Cronbach's Alpha |
|---|---|---|---|---|
| ACCS | | | | |
| ACCS3 | .813 | 0.933 | 0.822 | .919 |
| ACCS2 | .762 | | | |
| ACCS1 | .652 | | | |
| SPES | | | | |
| SPES1 | .799 | 0.929 | 0.813 | .919 |
| SPES2 | .790 | | | |
| SPES3 | .603 | | | |
| CONI | | | | |
| CONI3 | 1.239 | 0.744 | 0.523 | .794 |
| CONI2 | .179 | | | |
| CONI1 | .217 | | | |
| SERQ | | | | |
| SERQ1 | .597 | 0.877 | 0.704 | .875 |
| SERQ2 | .796 | | | |
| SERQ3 | .717 | | | |
| SYSQ | | | | |
| SYSQ1 | .596 | 0.879 | 0.708 | .876 |
| SYSQ2 | .838 | | | |
| SYSQ3 | .690 | | | |
| INFQ | | | | |
| INFQ4 | .507 | 0.765 | 0.619 | .884 |
| INFQ1 | .616 | | | |
| INFQ2 | .775 | | | |
| INFQ3 | .585 | | | |

Source: Author's Work

0.079, and Standardized Root Mean Square Residual (SRMR) of 0.0550. These values collectively indicate that the measurement model in this study demonstrates both scale reliability and validity, thus providing a robust foundation for testing the proposed hypotheses.

## 5.3 Hypotheses testing

In this study, the Structural Equation Model (SEM) was employed using SPSS AMOS to analyze each construct and their interrelationships, depicted in the structural model in Fig 3. SEM is widely recognized for its quantitative hypothesis testing capabilities [101, 102]. A series of goodness-of-fit indices, consistent with the methodology outlined by Schumacker and Lomax [103], were employed to assess the adequacy of the structural model. The obtained values for these indices include a χ2/d ratio of 8.339, Goodness of Fit Index (GFI) of .880, Adjusted Goodness of Fit Index (AGFI) of .832, Parsimonious Goodness of Fit Index (PGFI) of .630, Comparative Fit Index (CFI) of .948, Tucker-Lewis Index (TLI) of .934, and Root Mean Square Error of Approximation (RMSEA) of .082. These values collectively suggest that the structural model presented in Fig 3 fits well with the sample data. Furthermore, all hypotheses were assessed using SEM with Latent Variables (LV) based on the maximum likelihood estimation method, with the results summarized in Table 6. Notably, all hypotheses except H1a, H3a, and H7 were found to be supported by the data.

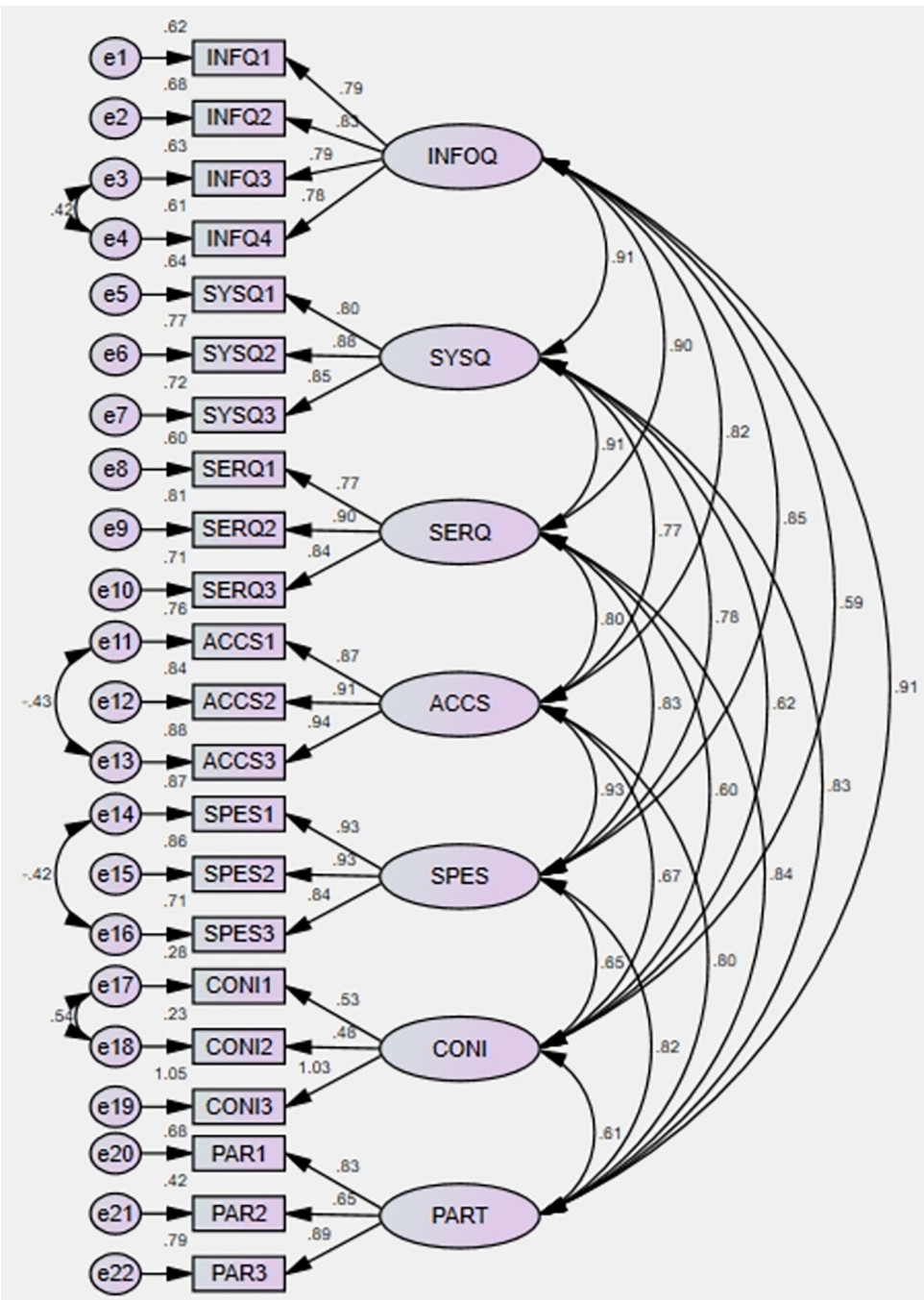

**Fig 2. Measurement model.**

## 5.4 Mediation testing

In this study, the mediation effects of SPES and ACCS on various causal relationships were analyzed by following the framework proposed by Baron and Kenny [104]. The results, summarized in Table 7, indicate significant mediating influences of SPES on the relationships between SERQ and ACCS, SYSQ and ACCS, and INFQ and ACCS. Besides, ACCS significantly mediate the relationship between SPES and CONI. Specifically, the p-values for the

**Table 5. Discriminant validity and correlation matrix.**

|  | CR | AVE | MSV | MaxR(H) | CONI | INFOQ | SYSQ | SERQ | ACCS | SPES | CITP |
|---|---|---|---|---|---|---|---|---|---|---|---|
| CONI | 0.744 | 0.523 | 0.449 | 1.057 | **0.723** |  |  |  |  |  |  |
| INFOQ | 0.765 | 0.619 | 0.826 | 0.765 | 0.593 | **0.787** |  |  |  |  |  |
| SYSQ | 0.879 | 0.708 | 0.823 | 0.884 | 0.616 | 0.909 | **0.841** |  |  |  |  |
| SERQ | 0.877 | 0.704 | 0.823 | 0.890 | 0.599 | 0.904 | 0.907 | **0.839** |  |  |  |
| ACCS | 0.933 | 0.822 | 0.865 | 0.938 | 0.670 | 0.823 | 0.766 | 0.800 | **0.907** |  |  |
| SPES | 0.929 | 0.813 | 0.865 | 0.938 | 0.655 | 0.846 | 0.780 | 0.835 | 0.930 | **0.902** |  |
| CITP | 0.835 | 0.631 | 0.826 | 0.869 | 0.613 | 0.909 | 0.832 | 0.844 | 0.799 | 0.818 | **0.794** |

Note(s): Model fit indices: $\chi^2/d$ = 3.07, GFI = 0.864, AGFI = 0.812, CFI = 0.943, TLI = 0 .928, NFI .918. IFI = 0.943 RMSEA = 0.079, SRMR = 0.0550. Bold diagonal values are the square root of AVEs.

Source(s): Authors work

mediation effects are as follows: SERQ -> SPES -> ACCS is 0.002, SYSQ -> SPES -> ACCS is 0.024, INFQ -> SPES -> ACCS is 0.001, and SPES -> ACCS -> CONI is 0.003. Consequently, hypotheses H9a, H9b, H9c, and H9d, which propose these mediation effects, are supported by the data. This suggests that SPES plays a significant mediating role in shaping attitudes towards accumulative satisfactions in the context of the relationships between service quality, system quality, information quality, and customer use intention of the services offered by the PDC.

## 5.5 Moderation testing

In this study, moderation analysis was conducted using SPSS PROCESS v4.2 by Hayes [105], with SPSS version 20.0. A confidence level of 95% and 5000 bootstrap samples were utilized for all confidence intervals. Conditional tables were generated using the 16th, 50th, and 84th percentiles. Furthermore, moderated mediation effects of CITP (Citizen Participation) on the relationships between different constructs were computed based on the principles outlined by Bollen [106]. The results, presented in Table 8 and Fig 4, indicate that CITP significantly moderates mediation effects. Specifically, the Index of moderated mediation for CITP on the

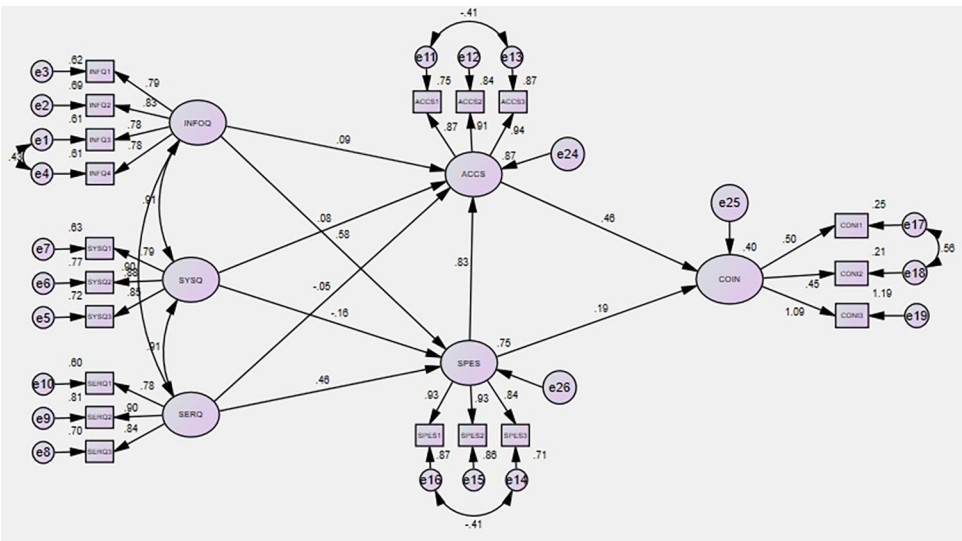

**Fig 3. Structural model.**

**Table 6. Hypothesis results.**

| Hypotheses | Relationship | Estimate | S.E. | T-value | P-value | Comments |
|---|---|---|---|---|---|---|
| H1a | INFQ -> ACCS | .057 | .047 | 1.218 | .223 | Reject |
| H1b | INFQ -> SPES | .452 | .047 | 9.639 | *** | Accept |
| H2a | SYSQ -> ACCS | .072 | .029 | 2.495 | .013 | Accept |
| H2b | SYSQ -> SPES | .122 | .032 | 3.752 | *** | Accept |
| H3a | SERQ -> ACCS | .011 | .044 | .242 | .809 | Reject |
| H3b | SERQ -> SPES | .421 | .043 | 9.805 | *** | Accept |
| H6 | SPES -> ACCS | .923 | .089 | 10.353 | *** | Accept |
| H7 | SPES -> COIN | .119 | .084 | 1.403 | .161 | Reject |
| H8 | ACCS -> COIN | .261 | .090 | 2.904 | .004 | Accept |

Note(s): Model fit indices: $\chi^2/d$ = 8.339, GFI = .880, AGFI = .832, PGFI = .630, CFI = .948, TLI = .934, RMSEA = .082

Source(s): Authors work

relationships between INFQ and ACCS, SYSQ and ACCS, SERQ and ACCS, INFQ and SPES, SYSQ and SPES, and SERQ and SPES were computed. The Index of moderated mediation values and their 95% confidence intervals are as follows: INFQ and ACCS: index = -.0058, 95% CI = [-.0324; .0226]; SYSQ and ACCS: index = -.0045, 95% CI = [-.0379; .0302]; SERQ and ACCS: index = -.0084, 95% CI = [-.0417; .0196]; INFQ and SPES: index = -.0156, 95% CI = [-.0439; .0040]; SYSQ and SPES: index = -.0200, 95% CI = [-.0516; .0079]; SERQ and SPES: index = -.0167, 95% CI = [-.0482; .0086]. These results provide strong support for the hypotheses no. H4a, H4b, H4c, H5a, H5b, and H5c, suggesting that CITP moderates the mediation effects between the specified constructs as indicated.

## 6. Discussion

This study intended to assess the service quality, satisfaction and intention to use the PDC, a digital service center at Pourasava in Bangladesh. To address the objective, this study used the D&M IS success model and Zhang's two-dimensional satisfaction model, which is prominent for the e-service quality satisfaction assessment. To meet the study objective, this study set 9 hypotheses based on the above two models to be tested. The study results supported all but H1a, H3a, and H7.

Surprisingly, this study rejected the first hypothesis (H1a) that no significant relationship was found between INFQ and ACCS (as a result of the P-value being .223), which contradicts the findings of the previous study of Hossain et al. [84], Biswas and Roy, [15]; and Ming et al.

**Table 7. Mediation results.**

| Indirect Paths | Unstandardized Estimate | Lower Bound | Upper Bound | P-value | Standardized Estimate | Comment |
|---|---|---|---|---|---|---|
| SERQ->SPES->ACCS | 0.388 | 0.265 | 0.510 | 0.002 | 0.430** | Mediate |
| SYSQ->SPES->ACCS | 0.112 | 0.037 | 0.203 | 0.024 | 0.140* | Mediate |
| INFQ->SPES->ACCS | 0.147 | 0.303 | 0.563 | 0.001 | 0.447*** | Mediate |
| SPES->ACCS->CONI | 0.240 | 0.135 | 0.387 | 0.003 | 0.327** | Mediate |

Note(s)

*** p<0.001

** p<0.010

* p<0.050 and Confidence Interval 95%

Source(s): Authors work

**Table 8. Results of moderated mediation analysis.**

| Variable | | Mediator | Effect | BootSE | BootLLCI | BootULCI | Result |
|---|---|---|---|---|---|---|---|
| ACCS | INFQ | Low level of CITP | .0226 | .0520 | -.0815 | .1218 | Significant |
| | | High level of CITP | .0177 | .0409 | -.0638 | .0973 | |
| | | Index of Moderated Mediation | -.0058 | .0139 | -.0324 | .0226 | |
| | SYSQ | Low level of CITP | .0115 | .0436 | -.0738 | .0961 | Significant |
| | | High level of CITP | .0077 | .0293 | -.0507 | .0649 | |
| | | Index of Moderated Mediation | -.0045 | .0174 | -.0379 | .0302 | |
| | SERQ | Low level of CITP | .0270 | .0455 | -.0636 | .1190 | Significant |
| | | High level of CITP | .0199 | .0336 | -.0475 | .0888 | |
| | | Index of Moderated Mediation | -.0084 | .0151 | -.0417 | .0196 | |
| SPES | INFQ | Low level of CITP | .0941 | .0571 | -.0227 | .2029 | Significant |
| | | High level of CITP | .0809 | .0496 | -.0191 | .1752 | |
| | | Index of Moderated Mediation | -.0156 | .0121 | -.0439 | .0040 | |
| | SYSQ | Low level of CITP | .0655 | .0483 | -.0268 | .1628 | Significant |
| | | High level of CITP | .0487 | .0365 | -.0198 | .1232 | |
| | | Index of Moderated Mediation | -.0200 | .0153 | -.0516 | .0079 | |
| | SERQ | Low level of CITP | .0711 | .0550 | -.0363 | .1760 | Significant |
| | | High level of CITP | .0571 | .0447 | -.0281 | .1458 | |
| | | Index of Moderated Mediation | -.0167 | .0146 | -.0482 | .0086 | |

Note(s): SE = standard error, LLCI = low level of confidence interval, ULCI = upper level of confidence interval, INFQ = Information Quality, SYSQ = System Quality, SERQ = Service Quality, CITP = Citizen Participation.

Source: Author's Work

[77]. This study found no positive relationship between INFQ and ACCS but a significant association between INFQ and SPES (as the P-value is .00) and supported the H1b. In this regard, this study agreed with the previous findings of Hossain et al. [84], Biswas and Roy [15], and Ming et al. [77]. Therefore, this study argued that information quality can influence specific satisfaction, which might not lead to accumulative satisfaction. The results of the study also supported the H2a and H2b that a significant positive relationship was found in both cases: between SYSQ and ACCS (P-value is .013) and between SYSQ and SPES (P-value is .00), which advocate the studies of the past researchers, i.e. Hossain et al, [84]; Ferdous et al, [51]; Alam, [50]; Biswas and Roy [15]; and Yanjun, [64]. As a result, this study claimed that citizen satisfaction depends on the system quality.

The study result shows that there is no positive impact of SERQ on the ACCS (as the P-value is .809) and rejected the H3a, which opposes the findings of the previous studies, i.e. Saha [46]; Biswas and Roy [15]; Hoque, [47]; Rita et al., (2019) and Morgeson (2014). On the contrary, the result found a significant influence of SERQ on the SPES (as the P-value is .00) and accepted the H3b, which provides a similar view to previous studies of Saha (2022); Biswas and Roy (2020); Hoque (2020); and Rita et al. [40]. Thus, this study opined that service quality indeed affects citizen-specific satisfaction, but it does not mean affecting accumulative satisfaction.

This study computed moderated mediation analysis to determine how CITP mediates the relationships among the constructs. The analysis results supported the related hypotheses, i.e. H4a, b, c, and H5a, b, c, as they found it significant. The CITP has a great influence on moderating the relationship between INFQ and ACCS, INFQ and SPES, SYSQ and ACCS, SYSQ and SPES, SERQ and ACCS; and SERQ and SPES since the P-value of all the construct's relations is 0.00. These findings support past studies like Hossain et al. [84], Claycomb et al. [65],

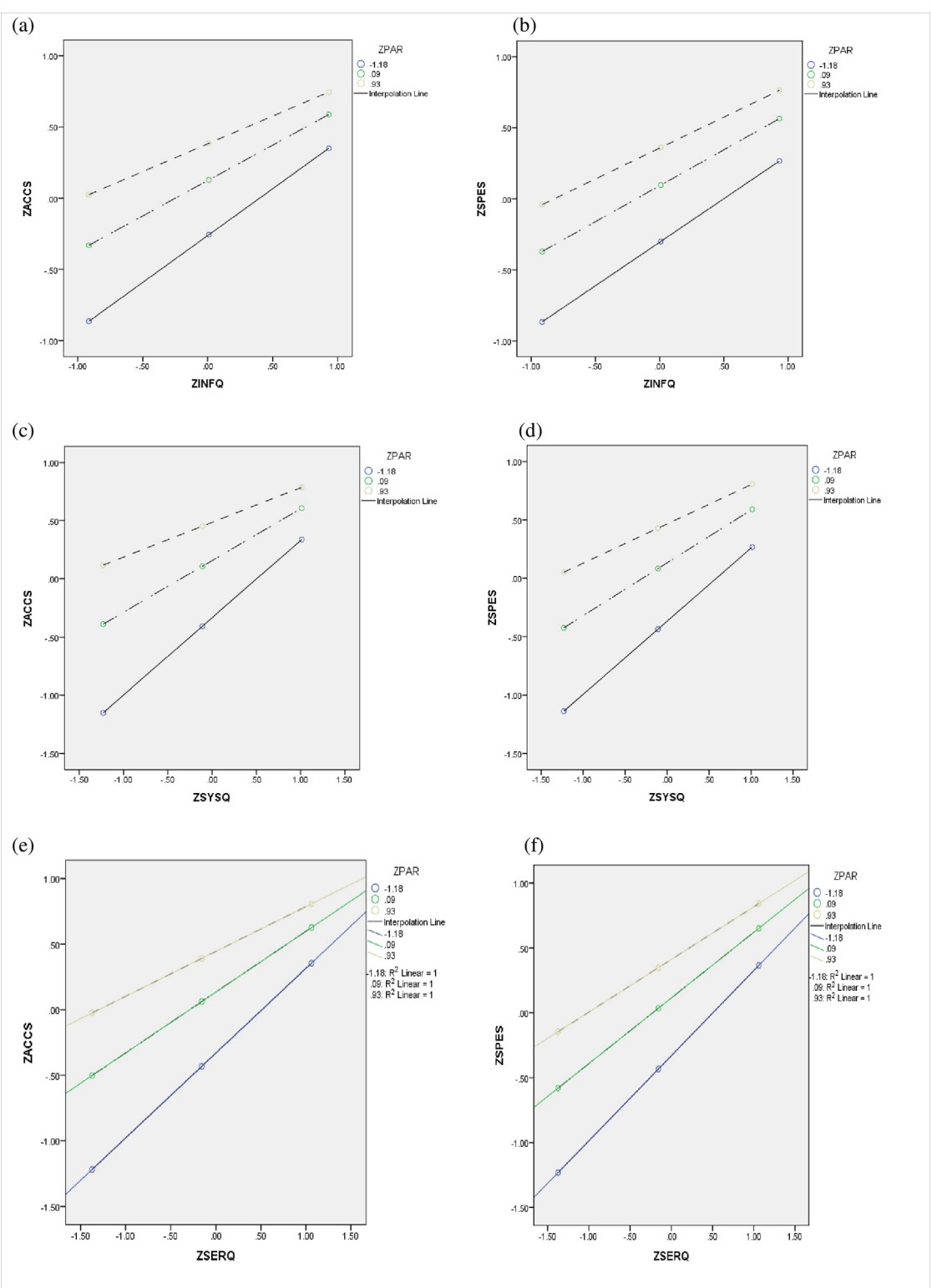

**Fig 4. Moderation results.**

Khadiza and Nurullah [48], Beatson et al. [33] and Biswas and Roy [15]. Therefore, this study argued that citizen participation significantly moderates the quality and satisfaction relationship.

To find the relationship between the two dimensions of satisfaction given by Zhang [58], this study set hypothesis (H6) that SPES has a significant positive impact on the ACCS. The study result found a positive relationship between them and accepted the H6 (as the p-value is .00) and supported the findings of the previous studies, i.e. Alsuwaidi & Sultan [45]; Ferdous et al. [51]; Biswas and Roy [15]; Zhang [58]; and Caron and Giauque [37]. So, this study stated that specific citizen satisfaction significantly affects their accumulative satisfaction; hence, accumulative happiness largely depends on the particular pleasure.

This study also observed the relationship between satisfaction and continuous use intention of the citizens and tested the hypotheses, i.e. H7 and H8. The results, unfortunately, found no significant impacts of the SPES on the USEI of the citizen and rejected the H7 (as the p-value is .161), which contradicts the previous studies' results of Saha [46] and Anam [49]. However, this study, on the other hand, found a significant positive impact of the ACCS on the USEI of the citizen (as the p-value is .004) and supported the H8; it accepts the past studies' findings of Biswas and Roy [15]; and Chatfield & AlAnazi [39]. Thus, this study argued that the continuous use intention of the citizen is not based on specific satisfaction but significantly depends on active satisfaction.

This study also computed the mediator analysis of how SPES mediates the relationship between SERQ and ACCS; SYSQ and ACCS; and INFQ ACCS, and ACCS mediates the relationship between SPES and CONI. The results found significant positive mediating effects on all the connections mentioned above as the p-value for all is highly substantial and accepted the H9a, H9b, H9c and H9d, which supports the previous studies like Rahman [86]; Abdirad & Krishnan, [107]; Preaux et al., [87]. Therefore, this study claimed that specific satisfaction mediates the relationship between quality and accumulative satisfaction, and accumulative satisfaction mediates the relationship between particular satisfaction and continuous intention to use the citizen.

## 7. Research implication

As the first comprehensive PDC research, this study has substantial theoretical and practical implications. The high acceptance of most set hypotheses underscores the reliability and validity of the findings, contributing significantly to the literature on e-governance service quality, satisfaction, and management within the context of PDCs in Bangladesh.

### 7.1 Theoretical implication

In developing the conceptual framework, this study has successfully integrated the D&M IS success model and the theory of citizen satisfaction. Providing a robust theoretical framework to understand the facts and figures of the e-service management system of PDC in Bangladesh, focusing on quality and satisfaction. This study agreed with the significant findings of the previous researchers, except a few ensure the high acceptability of the theoretical arguments that the survey made to the researcher, practitioners, and the citizen. The study's theoretical contributions highlighted that citizen service satisfaction is highly dependent on the quality ensured in the PDC, and quality in terms of information, service, and system attracts the citizen to use the PDC continuously. Therefore, this study has a solid theoretical implication, explaining the human behavior of service quality, satisfaction, and continuous intention to use. In addition, since this is the first rigorous quantitative research on the PDC, this study is a genuine direction for future researchers, paving who the way for conducting research in this area of PDC and contribute to national development through their research work.

## 7.2 Practical implication

The empirical findings of this study offer essential guidelines for the PDC authorities (concerned departments, ministries, or other bodies) to enhance its current service scenario so that they may offer a better service system to the community through which people will be satisfied, leading towards their continuous intention to use the PDC. To enhance the practical utility of PDC, concrete steps for its improvement based on the study's results are crucial. For instance, this study found that specific satisfaction largely depends on the information quality, system quality, and service quality, which attract the citizens to continue usage of PDC. To leverage this insight, PDC staff must be careful in ensuring the quality of PDC services, building trust among users for continued usage without any hesitation. Additionally, the study has found that citizen participation has a tremendous moderating effect on the links between the quality of PDC and citizen satisfaction. Therefore, the concerned authorities should actively work to facilitate active citizen participation by providing updated and authentic information, quality services and a better environment so that the citizens have a positive mindset to use the PDC again. This study will undoubtedly facilitate the government and the policymakers as it shows a well-designed theoretical framework and critical empirical findings to renovate the PDCs in Bangladesh for the future betterment of the community.

## 8. Conclusion

As worldwide growing changes in the service delivery and management through digitalization and ICT development, Bangladesh started its digitalization in almost all sectors at both national and local level. PDC is a one-stop service center of urban local government in Bangladesh and has become an essential service delivery institution getting popular among the citizens. It is meaningful to analyze the state of service quality, citizen satisfaction, and participation of PDC because of its growing popularity. This study designed a theoretical framework based on the D&M IS success model and Zhang's dimension satisfaction model for this investigation. This study delves into the complex correlation that exists between citizen participation and service quality, satisfaction, and intention to use Pourasava Digital Center in Bangladesh. Upon conducting a thorough investigation, it is apparent that the level of satisfaction and intention to use the services are strongly influenced by the service quality offered by the digital center. The research also emphasizes the moderating effect of citizen participation, showing that proactive citizen engagement amplifies the influence of service quality on customer satisfaction and intention to use digital services. The analysis shows that the PDC service quality; the system quality they established, and the information quality they offer on their website heavily affect the satisfaction of those who consume the services. Active citizen participation through taking services, knowing information and giving suggestions signifies the PDC's service quality and satisfaction relation positively.

The findings of this research hold crucial implications for policymakers, administrators, and stakeholders involved in digital initiatives aimed at citizen service delivery. By prioritizing service quality enhancements and fostering citizen participation, authorities can effectively bolster satisfaction levels and encourage greater utilization of digital centers. Additionally, recognizing the nuanced interplay between these variables underscores the importance of implementing tailored strategies to optimize service delivery and foster citizen engagement.

Based on the findings, there are some important insights to improve overall condition of the PDC; In order to facilitate the provision of services in a hassle-free manner, the PDC authority ought to establish a flexible working schedule that ensures clients are aware of the duration of service delivery and entrepreneurs arrive at the PDC office on time. An appropriate training program for entrepreneurs should be set up, as training is essential for acclimating

new hires to the services and preparing them to become specialists in service delivery. Adding additional entrepreneurs to the service may also be a way to address the issue. In addition, periodic performance evaluations and assessments ought to be used to track and upgrade the caliber of customer care agents. Without consequences and incentives for their actions, the monitoring system alone will not function.

Overall, the empirical findings of this study could be instrumental and highly suggestive of making sustainable, trustworthy and effective PDCs in Bangladesh for uplifting the life standard of the community people.

### 8.1 Limitations and future research guidelines

Despite having immense theoretical and practical implications, particularly for being the first quantitative study on the PDC in Bangladesh, this study has some limitations; firstly, it was conducted on five selected PDCs with 332 samples, which could narrow the scope of generalization. So, this study allows future researchers to research the PDC with a larger sample size and broader context. Secondly, this study excluded some questionnaires with missing values that may add other important information. This study included 7 LV and 21 MV for this investigation, which might ignore some crucial variables like Citizen Trust. So, the future researcher could conduct their research by including them in the theoretical framework. Fourthly, this study used SEM to compute data where further data verification may be needed to ensure more accuracy of the result; therefore, future researchers may use updated tools like Artificial Neural Networks (ANN). Finally, this study was conducted in Bangladesh, where application may be limited; other developing countries could be considered for future research.

## Supporting information

**S1 Dataset.**
(SAV)

## Author Contributions

**Conceptualization:** Bikram Biswas.

**Data curation:** Md Mostafizur Rahman.

**Formal analysis:** Md Mostafizur Rahman.

**Investigation:** Anas Al Masud.

**Methodology:** Bikram Biswas.

**Resources:** Anas Al Masud.

**Supervision:** Bikram Biswas.

**Writing – original draft:** Mohammad Nur Ullah.

**Writing – review & editing:** Bikram Biswas.

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
