## [Decision Letter · Decision Letter 0]

2 Jan 2024

PONE-D-23-34691An Empirical study of Service Quality and Satisfaction of Pourasava Digital Center (PDC) in Bangladesh: The Moderating Effect of Citizen ParticipationPLOS ONE

Dear Dr. Biswas,

Thank you for submitting your manuscript to PLOS ONE. After careful consideration, we feel that it has merit but does not fully meet PLOS ONE’s publication criteria as it currently stands. Therefore, we invite you to submit a revised version of the manuscript that addresses the points raised during the review process.

We look forward to receiving your revised manuscript.

Kind regards,

Zhiyuan Yu

Academic Editor

PLOS ONE

Journal Requirements:

3. Please ensure that you include a title page within your main document. You should list all authors and all affiliations as per our author instructions and clearly indicate the corresponding author.

4. Please clarify the Table 6 "Table 6: Hypothesis Results" in page "13" and Table 6 "Table 6. Results of Moderated Mediation Analysis" in page "14".

5. Please upload a copy of Supporting Information Figure/Table/etc. S1 Fig-S4 Fig which you refer to in your text on page 23.

Additional Editor Comments:

Both the reviewers proposed some major concerns. Please carefully address those issues.

Reviewers' comments:

Reviewer's Responses to Questions

**Comments to the Author**

1. Is the manuscript technically sound, and do the data support the conclusions?

Reviewer #1: Yes

Reviewer #2: Yes

2. Has the statistical analysis been performed appropriately and rigorously? 

Reviewer #1: Yes

Reviewer #2: Yes

3. Have the authors made all data underlying the findings in their manuscript fully available?

Reviewer #1: Yes

Reviewer #2: Yes

4. Is the manuscript presented in an intelligible fashion and written in standard English?

Reviewer #1: Yes

Reviewer #2: No

5. Review Comments to the Author

Reviewer #1: In this paper, the author proposes a study of Service Quality and Satisfaction of Pourasava Digital Center (PDC) in Bangladesh. Overall, the paper demonstrates a solid understanding of the subject matter but could be improved by providing a clearer statement of the research problem in the introduction, a more critical analysis in the literature review, and a more detailed rationale for the hypotheses. These enhancements would contribute to a more robust and compelling research paper.

1. Introduction: The introduction sets the stage for the study but could benefit from a more detailed exposition of the research context. Specifically, the last paragraph lacks a clear articulation of the research gap that this study aims to address. Enhancing this section with a concise statement of the research problem and its significance would strengthen the foundation of the paper.

2. Literature Review: The literature review appears comprehensive but somewhat lacks critical analysis. The authors might consider not only summarizing existing studies but also critically evaluating them, highlighting contradictions or gaps in the literature. Additionally, providing clear explanations for the choice of constructs in the proposed model would be advantageous for the readers. This would better position their study within the existing body of knowledge and justify the need for their research.

3. Theoretical Framework and Hypotheses Development: The theoretical framework is well-constructed, integrating relevant theories effectively. However, the development of hypotheses seems to be more descriptive & brief than analytical & comprehensive. It would be beneficial to provide a more in-depth rationale for each hypothesis, linking them more explicitly to the theoretical framework. This would enhance the reader's understanding of how the hypotheses are grounded in the theory.

4. Statistical Analysis: The statistical analysis is comprehensive, but the interpretation of some model fit indices (e.g., RMSEA of .149) (mentioned in Section 5.3 Hypotheses Testing) suggests potential issues with model fit. A RMSEA value of .149 is generally considered high, indicating a poor fit. The authors should address this in their discussion, considering either model re-specification or providing a rationale for the acceptance of these fit indices. Additionally, the rationale behind accepting or rejecting specific hypotheses should be clearly articulated, especially in cases where the results are marginal or counterintuitive. The following information may be helpful for the authors:

RMSEA ≤ 0.05: Indicates a close fit of the model in relation to the degrees of freedom.

RMSEA between 0.05 and 0.08: Suggests a reasonable error of approximation in the population.

RMSEA between 0.08 and 0.10: Represents a mediocre fit.

RMSEA > 0.10: Is considered a poor fit.

5. Discussion: The discussion effectively connects study results with previous research, but it could benefit from deeper analysis of why certain hypotheses were not supported. Specifically, exploring potential reasons for the unexpected findings (e.g., H1a, H3a, H7) would provide a more comprehensive understanding and contribute to the field's knowledge.

6. Research Implications: The implications are broadly stated but lack specificity in how the findings can be practically applied. Detailing concrete steps for PDC improvement based on the study's results would enhance the practical utility of the research.

while the paper offers valuable insights into the Service Quality and Satisfaction at PDC in Bangladesh, refining its research problem articulation, deepening analytical rigor, and addressing statistical concerns will significantly enhance its scholarly and practical contributions.

Reviewer #2: Dear author although topic of research is interesting how need improvements for example scope of this study is limited to one country, why you do not add other countries studies conducted in same region and enhance scope of your research for example "Investigating the role of diffusion of innovation theory in determining citizen’s intention towards adoption of e-government services" this study is also conducted in south Asia context and relevant to discuss

Improvements are required in your methodology section information regarding research paradigm is missing it is highly recommended to design your research under a specific research paradigm and you may grab this information through "Research design and methods: A systematic review of research paradigms, sampling issues and instruments development" similarly support scale adoption with latest research "Integration of DeLone and McLean and self-determination theory in internet banking continuance intention context" for information and system quality scale.

you are requested to improve research conclusion it is too short and not justifiable

6. PLOS authors have the option to publish the peer review history of their article (what does this mean?). If published, this will include your full peer review and any attached files.

Reviewer #1: **Yes: **Dr. Imdadullah Hidayat-ur-Rehman

Reviewer #2: **Yes: **SAMAR RAHI

---

## [Author Response · Author response to Decision Letter 0]

19 Feb 2024

Dear concern, we have corrected our file based on your valuable comments. thank you

---

## [Editor Report · Decision Letter 1]

26 Feb 2024

PONE-D-23-34691R1An Empirical study of Service Quality and Satisfaction of Pourasava Digital Center (PDC) in Bangladesh: The Moderating Effect of Citizen ParticipationPLOS ONE

Dear Dr. Nur Ullah,

Thank you for submitting your manuscript to PLOS ONE. After careful consideration, we feel that it has merit but does not fully meet PLOS ONE’s publication criteria as it currently stands. Therefore, we invite you to submit a revised version of the manuscript that addresses the points raised during the review process.

We look forward to receiving your revised manuscript.

Kind regards,

Dr. Zhiyuan Yu

Academic Editor

PLOS ONE

Additional Editor Comments:

Before out for review, please carefully check the revised manuscript, which contains duplicate pieces for each document (total 213 pages are too large). Please remove the duplicate documents.

---

## [Author Response · Author response to Decision Letter 1]

28 Feb 2024

Dear Concern, We have corrected all the files according to academic editor and reviewer's comments. And the files are attached. Thank you

---

## [Decision Letter · Decision Letter 2]

22 Mar 2024

PONE-D-23-34691R2Service Quality, Satisfaction, and Intention to Use Pourasava Digital Center in Bangladesh: The Moderating Effect of Citizen ParticipationPLOS ONE

Dear Dr. Nur Ullah,

Thank you for submitting your manuscript to PLOS ONE. After careful consideration, we feel that it has merit but does not fully meet PLOS ONE’s publication criteria as it currently stands. Therefore, we invite you to submit a revised version of the manuscript that addresses the points raised during the review process.

We look forward to receiving your revised manuscript.

Kind regards,

Dr. Zhiyuan Yu

Academic Editor

PLOS ONE

Reviewers' comments:

Reviewer's Responses to Questions

**Comments to the Author**

1. If the authors have adequately addressed your comments raised in a previous round of review and you feel that this manuscript is now acceptable for publication, you may indicate that here to bypass the “Comments to the Author” section, enter your conflict of interest statement in the “Confidential to Editor” section, and submit your "Accept" recommendation.

Reviewer #1: All comments have been addressed

Reviewer #2: All comments have been addressed

2. Is the manuscript technically sound, and do the data support the conclusions?

Reviewer #1: Yes

Reviewer #2: No

3. Has the statistical analysis been performed appropriately and rigorously? 

Reviewer #1: Yes

Reviewer #2: No

4. Have the authors made all data underlying the findings in their manuscript fully available?

Reviewer #1: Yes

Reviewer #2: No

5. Is the manuscript presented in an intelligible fashion and written in standard English?

Reviewer #1: Yes

Reviewer #2: No

6. Review Comments to the Author

Reviewer #1: (No Response)

Reviewer #2: Dear authors unfortunately you have not address issues raised by reviewer research paradigm information is still missing and even article is not structured accordingly

"Reviewer #2: Dear author although topic of research is interesting how need

improvements for example scope of this study is limited to one country, why you do not

add other countries studies conducted in same region and enhance scope of your

research for example "Investigating the role of diffusion of innovation theory in

determining citizen’s intention towards adoption of e-government services" this study is

also conducted in south Asia context and relevant to discuss

Improvements are required in your methodology section information regarding research

paradigm is missing it is highly recommended to design your research under a specific

research paradigm and you may grab this information through "Research design and

methods: A systematic review of research paradigms, sampling issues and instruments

development" similarly support scale adoption with latest research "Integration of

DeLone and McLean and self-determination theory in internet banking continuance

intention context" for information and system quality scale. you are requested to

improve research conclusion it is too short and not justifiable"

7. PLOS authors have the option to publish the peer review history of their article (what does this mean?). If published, this will include your full peer review and any attached files.

Reviewer #1: **Yes: **Dr Imdadullah Hidayat-ur-Rehman

Reviewer #2: **Yes: **Samar Rahi

---

## [Author Response · Author response to Decision Letter 2]

29 Apr 2024

Responses to Reviewer and Academic Editor 

Comments to the Author

1. If the authors have adequately addressed your comments raised in a previous round of review and you feel that this manuscript is now acceptable for publication, you may indicate that here to bypass the “Comments to the Author” section, enter your conflict of interest statement in the “Confidential to Editor” section, and submit your "Accept" recommendation.

Reviewer #1: All comments have been addressed

Reviewer #2: All comments have been addressed

Authors’ justification Reviewer #1 & #2: Thank you for your comments that gave us confidence, we have tried our best. 

2. Is the manuscript technically sound, and do the data support the conclusions?

Reviewer #1: Yes

Authors’ justification Reviewer #1: Thank you so much for your appreciating response that gave us confidence, we have tried our best. 

Reviewer #2: No

Authors’ justification Reviewer #2: Thank you so much for your response in this regard that helped us to improve our manuscript regarding making a strong linkage between data and conclusion of the manuscript. We have updated according to your suggestions you made here. 

3. Has the statistical analysis been performed appropriately and rigorously?

Reviewer #1: Yes

Authors’ justification Reviewer #1: Thank you so much for your appreciating response that gave us confidence, we have tried our best. 

Reviewer #2: No

Authors’ justification Reviewer #2: Thank you so much for your response. We think we applied the correct statistical tool to analyze the data and tried best to analyze them in appropriate manner. 

4. Have the authors made all data underlying the findings in their manuscript fully available?

Reviewer #1: Yes

Authors’ justification to Reviewer #1: Thank you for your comment, we already made all the data used in this study available. 

Reviewer #2: No

Authors’ justification to Reviewer #2: Thank you for your comment, but we have published all the data used in this study to the journal through submitting data file as supporting documents. 

5. Is the manuscript presented in an intelligible fashion and written in Standard English?

Reviewer #1: Yes

Authors’ justification to Reviewer #1: Thank you for your comment. 

Reviewer #2: No

Authors’ justification to Reviewer #2: Thank you for your comment that helped us to look over our manuscript and we gone through our paper and updated English language, after that the manuscript is revised by an English language expert for ensuring Standard English. 

6. Review Comments to the Author

Reviewer #1: (No Response)

Reviewer #2: Dear authors unfortunately you have not address issues raised by reviewer research paradigm information is still missing and even article is not structured accordingly

Author’s Response to Reviewer #2: Thank you so much for your feedback. We have updated our manuscript based on your feedback. 

"Reviewer #2: Dear author although topic of research is interesting how need

improvements for example scope of this study is limited to one country, why you do not

add other countries studies conducted in same region and enhance scope of your

research for example "Investigating the role of diffusion of innovation theory in

determining citizen’s intention towards adoption of e-government services" this study is

also conducted in south Asia context and relevant to discuss

Improvements are required in your methodology section information regarding research

paradigm is missing it is highly recommended to design your research under a specific

research paradigm and you may grab this information through "Research design and

methods: A systematic review of research paradigms, sampling issues and instruments

development" similarly support scale adoption with latest research "Integration of

DeLone and McLean and self-determination theory in internet banking continuance

intention context" for information and system quality scale. You are requested to

improve research conclusion it is too short and not justifiable"

Author’s Response to Reviewer #2: Thank you so much for your rigorous feedback which helped us to improve our manuscript scientifically and theoretically sound. We have updated our manuscript based on your comments.

---

## [Decision Letter · Decision Letter 3]

8 May 2024

Service Quality, Satisfaction, and Intention to Use Pourasava Digital Center in Bangladesh: The Moderating Effect of Citizen Participation

PONE-D-23-34691R3

Dear Dr. Mohammad Nur Ullah,

We’re pleased to inform you that your manuscript has been judged scientifically suitable for publication and will be formally accepted for publication once it meets all outstanding technical requirements.

Kind regards,

Dr. Zhiyuan Yu

Academic Editor

PLOS ONE

Additional Editor Comments (optional):

Reviewers' comments:

Reviewer's Responses to Questions

**Comments to the Author**

1. If the authors have adequately addressed your comments raised in a previous round of review and you feel that this manuscript is now acceptable for publication, you may indicate that here to bypass the “Comments to the Author” section, enter your conflict of interest statement in the “Confidential to Editor” section, and submit your "Accept" recommendation.

Reviewer #2: All comments have been addressed

2. Is the manuscript technically sound, and do the data support the conclusions?

Reviewer #2: Yes

3. Has the statistical analysis been performed appropriately and rigorously? 

Reviewer #2: Yes

4. Have the authors made all data underlying the findings in their manuscript fully available?

Reviewer #2: Yes

5. Is the manuscript presented in an intelligible fashion and written in standard English?

Reviewer #2: Yes

6. Review Comments to the Author

Reviewer #2: Current changes are acceptable, author has shown efforts to improve manuscript, overall research methods are now clear, research objectives are also satisfactory, overall work is satisfactory and comprise quality contents

7. PLOS authors have the option to publish the peer review history of their article (what does this mean?). If published, this will include your full peer review and any attached files.

Reviewer #2: **Yes: **Dr. Samar Rahi

---

## [Editor Report · Acceptance letter]

17 May 2024

PONE-D-23-34691R3 

PLOS ONE

Dear Dr. Nur Ullah, 

I'm pleased to inform you that your manuscript has been deemed suitable for publication in PLOS ONE. Congratulations! Your manuscript is now being handed over to our production team.

Kind regards, 

on behalf of

Dr. Zhiyuan Yu 

Academic Editor

PLOS ONE